# Relations between Structure/Composition and Mechanics in Osteoarthritic Regenerated Articular Tissue: A Machine Learning Approach

**DOI:** 10.3390/ijms241713374

**Published:** 2023-08-29

**Authors:** Matteo Berni, Francesca Veronesi, Milena Fini, Gianluca Giavaresi, Gregorio Marchiori

**Affiliations:** 1Medical Technology Laboratory, IRCCS Istituto Ortopedico Rizzoli, Via Di Barbiano 1/10, 40136 Bologna, Italy; matteo.berni@ior.it; 2Surgical Sciences and Technologies, IRCCS Istituto Ortopedico Rizzoli, Via Di Barbiano 1/10, 40136 Bologna, Italy; gianluca.giavaresi@ior.it (G.G.); gregorio.marchiori@ior.it (G.M.); 3Scientific Direction, IRCCS Istituto Ortopedico Rizzoli, Via Di Barbiano 1/10, 40136 Bologna, Italy; milena.fini@ior.it

**Keywords:** large animal OA model, biomechanics, machine learning, orthobiologics

## Abstract

In the context of a large animal model of early osteoarthritis (OA) treated by orthobiologics, the purpose of this study was to reveal relations between articular tissues structure/composition and cartilage viscoelasticity. Twenty-four sheep, with induced knee OA, were treated by mesenchymal stem cells in various preparations—adipose-derived mesenchymal stem cells (ADSCs), stromal vascular fraction (SVF), and amniotic endothelial cells (AECs)—and euthanized at 3 or 6 months to evaluate the (i) biochemistry of synovial fluid; (ii) histology, immunohistochemistry, and histomorphometry of articular cartilage; and (iii) viscoelasticity of articular cartilage. After performing an initial analysis to evaluate the correlation and multicollinearity between the investigated variables, this study used machine learning (ML) models—Variable Selection Using Random Forests (VSURF) and Extreme Gradient Boosting (XGB)—to classify variables according to their importance and employ them for interpretation and prediction. The experimental setup revealed a potential relation between cartilage elastic modulus and cartilage thickness (CT), synovial fluid interleukin 6 (IL6), and prostaglandin E2 (PGE2), and between cartilage relaxation time and CT and PGE2. SVF treatment was the only limit on the deleterious OA effect on cartilage viscoelastic properties. This work provides indications to future studies aiming to highlight these and other relationships and focusing on advanced regeneration targets.

## 1. Introduction

Osteoarthritis (OA) is the most common degenerative pathology affecting human and animal joints, especially the knee [1], and its worldwide prevalence is increasing [2].

Although OA is today considered to be a whole joint disease, hyaline articular cartilage (AC) is entangled from the outset by a combination of degenerative and inflammatory phenomena, impairing cell signaling pathways [3]. Indeed, the primary sign of early OA is represented by degradation of the cartilage extracellular matrix (ECM), which activates the release of inflammatory cytokines and catabolic mediators, leading to a prevalence of catabolic activity on the anabolic mediator [4].

Therefore, no therapeutic infiltrative strategy has been able to fully counteract early OA processes once underway. Among the most recommended non-surgical treatments, we mention the use of lateral wedge insoles, supervised exercise, unsupervised exercise, aquatic exercise, education programs, topical or oral NSAIDs, and oral acetaminophen or narcotics [5]. Although infiltrative conservative treatments are mainly effective in the earlier stages of OA, they provide only temporary benefits and do not repair cartilage once degradation has started [6,7].

Therefore, surgery still remains the reference treatment, often leading to sub-optimal outcomes due to post-operative complications and not ensuring a full recovery of the joint mobility [8]. Thus, this approach is only viable in the late stages of OA.

In an effort to find a therapeutic, minimally invasive treatment able to counteract degeneration of AC during the early stages of OA, mesenchymal stem cells (MSCs) have emerged in recent decades as a promising solution [9].

MSCs can be easily harvested from a variety of tissues, showing multi-lineage differentiation, self-renewal, and immunomodulatory abilities, and trophic activity by releasing growth factors and anti-inflammatory cytokines that inhibit cell apoptosis and stimulate cell proliferation and angiogenesis, thereby promoting tissue regeneration [10,11].

Many preclinical and clinical studies have highlighted encouraging results in treating OA by MSCs [12]. Nevertheless, assessing which cell line represents the best solution, along with a complete understanding of MSCs’ mechanism of action, remains controversial [13]. For this reason, pre-clinical studies are still needed.

In pre-clinical studies focusing on OA, large animal models are usually preferred, mainly because (i) the anatomy of their joints is more similar, compared to small animal models, to that of humans, and (ii) they enable the extraction and analysis of synovial fluid [14]. From this perspective, and taking into consideration anatomy, size, and nature, as well as the relatively lower associated costs and rising popularity, the sheep represents an excellent model for such studies [15].

Considering the suitability of MSCs for the treatment of OA, few studies have investigated their potential efficacy, mainly, but not exclusively, focusing on in vivo large animal models, such as sheep [16,17,18]. In particular, a large animal model was used in our previous pre-clinical in vivo studies to compare three different MSC-based approaches in treating early knee OA [17,18]. Specifically, the therapeutic effects of adipose-derived mesenchymal stem cells (ADSCs), stromal vascular fraction (SVF), and amniotic endothelial cells (AECs) were evaluated, and encouraging results were obtained in terms of AC gross evaluation and mechanical response, after 3 months following injection [18].

When aiming to properly evaluate the impact of OA on the homeostasis of AC, the assessment of its biomechanics is fundamental. The onset and progression of OA alter the mechanics of AC [19]; moreover, the onset of this disease is often induced by mechanical over-stresses [20]. An integrated approach investigating the impact of OA on the AC macro- and micro-features must therefore consider the relation existing between tissue structure/composition (bio-) and function (-mechanics) to properly understand cell–tissue interaction [21]. It has been shown that (i) proteoglycan (PG) and collagen content correlate with cartilage viscosity; (ii) PG content and collagen orientation correlate with fibril elastic modulus; and (iii) content and orientation of the AC ECM change during OA progression [22].

Considering the pathogenesis of OA, synovial fluid inflammatory factors, e.g., Interleukin 1β (IL1β), IL6, Tumor Necrosis Factor α (TNFα), Cross Linked C-Telopeptide Of Type II Collagen (CTx2), and Prostaglandin E2 (PGE2) [23]; tissue composition and appearance, e.g., type II collagen content [24], fibrillation index [25], and tissue thickness [26]; and AC viscoelasticity, e.g., Young’s modulus, E, and relaxation time, τ [27,28,29], are just some of the features deregulated and impaired by the onset and progression of such a disease.

Therefore, the purpose of the present study was to reveal existing relations between articular tissue structure/composition—investigated by biochemistry and histology—and mechanics—assessed by indentation—in the case of various orthobiological treatments responding differently to the progression of such a disease. The evaluations were performed 3 and 6 months after an intra-articular injection of ADSCs, AECs, or SVF in the same OA in vivo model used in the previous study [17,18]. Data were analyzed using a machine learning (ML) approach to establish a prediction model that combines various histological and biochemical factors related to OA, relating the mechanical response of AC (Young’s modulus and relaxation time) to insights about the pathophysiology of the joint.

The combination of ML and biomedical research has resulted in significant advancement in the field of human health via the utilization of expansive biomedical data and tackling the growing complexity of biological systems, thus surpassing the limitations of traditional research methodologies. This has transformed the analysis and interpretation of intricate biological data, leading to the detection of biomarkers, the prediction of diseases, and the customization of therapies [30]. Biomedical analysis traditionally uses group comparison tests and regression for predictions. Assumptions of normal distribution and homogeneity of variance might not hold, leading to errors and oversight of complex variable interactions, thus impacting accuracy [31]. ML regression models, such as the regression tree model, are a predictive modeling technique based on decision tree structures and aim to predict a continuous numeric value as the output. The input data are split into subsets based on different features and their values in a regression tree. Aiming to minimize the variability of the target variable within each segment, the algorithm recursively partitions the data into smaller segments. For each step, the algorithm selects the feature and the corresponding value that best splits the data, typically based on criteria such as mean squared error or variance reduction. The result is a tree-like structure in which each leaf node represents a predicted numeric value. To predict a new data point, the tree follows the path along the input feature values, and the final output is the prediction at the leaf node [30].

## 2. Results

Preliminary data inspection highlighted that the two dependent variables—E and τ—presented univariate normality (Q-Q plots) and homogeneity of variance (residual fitted value plots), except for E, whose values were not normally distributed and were natural-log transformed (Shapiro–Wilk test: W = 0.96, *p* = 0.10). No significant correlations among dependent mechanical variables and independent variables were observed, but a significant correlation was found between lnE and τ (rs = −0.65, *p* < 0.0005) (Appendix A, Figure A1). Furthermore, various multicollinearities were found between continuous (histomorphometric and synovial fluid) and ordinal (treatment and macroscopic score (SD)) independent variables, which could cause problems in fitting the regression models and interpreting the results (Figure A1). No significant interactions or effects of treatment and time factors were found on mechanical AC lnE (Figure 1A), while for τ results, a significant decrease over time was found (8.2%, *p* = 0.011), which, moreover, remained slightly higher in AEC- and SVF-treated AC (Figure 1B).

To understand the relationship between independent and dependent (i.e., mechanical parameters) variables, the multicollinearity issue was addressed because the correlated variables will compete in explaining the dependent ones. For this reason, two non-parametric algorithms, VSURF and XGBoost, based on tree classification and splitting the importance of correlated independent variables, were applied to data in addition to linear models to define regression models describing AC mechanical parameters, i.e., lnE and τ. Subsequently, the datasets of the selected dependent and independent variables were divided into training and testing sets using an 80:20 split and a cross-validation of multivariate linear (MLR) and XGB regression (XGBR) models was performed. More details about the analysis conducted and the results obtained are given in Appendix A.

For each mechanical parameter, the validated MLR model had CT as a unique validated predictor; intermediate R^2^ values were found for τ (0.52), and the only reliable model obtained for lnE had an R^2^ value of almost zero. Table 1 reports the estimations and effectiveness of MLR, VSURF, and XGBR models for tested mechanical parameters and selected predictors; no significant differences were observed among MAE ± SDe of each model, highlighting that each method did not perform better than the others to obtain the best estimation effect.

Figure 2 shows the scatter plots of predictive versus actual values of lnE and τ according to VSURF regression models.

## 3. Discussion

The primary aim of this investigation was to elucidate relationships between AC functionality, i.e., viscoelastic properties, and knee pathophysiology, i.e., AC composition and physiology, and synovial fluid inflammatory factors, in a large animal model of early OA. This pathology was induced by lateral meniscectomy, and then addressed through various orthobiological MSC-based treatments. The performed analyses, based on ML, highlighted that AC mechanical features, i.e., lnE and τ, can be predicted starting from inflammatory factors of synovial fluid (i.e., IL6 and PGE2) and tissue histological features (i.e., CT) with a degree of effectiveness (R^2^) above 0.8. In addition, SVF was found to be a promising treatment in reducing the impact of OA on AC mechanical properties, i.e., by limiting the decrease in the investigated mechanical parameters between the two experimental times.

AC plays a key role in withstanding body loads, thanks to its heterogeneous structure and, consequently, complex mechanical response. Onset and progression of pathologies such as OA affect AC mechanical response [32]. Osteoarthritic AC is characterized by changes in thickness, mechanical properties, and permeability [33]. In early OA, the AC matrix presents a hypertrophic swelling [34], resulting in an increased thickness; further progression of OA leads to reduced structural integrity which, combined with increased wear, strongly reduces AC volume and, consequently, thickness [35]. Here, the link between CT and viscoelasticity was confirmed. In more detail, CT is present in both the models predicting lnE and τ values starting from biological and histological insights.

Focusing on the extent of the mechanical properties of AC reported here, Young’s modulus values agree with previous studies concerning the same animal model [36,37]. Regarding τ, no study has investigated its extent in the same animal model and by applying the same testing method. Although τ can be highly variable and dependent on the site, articulation, model from which AC is retrieved, and testing method [27,38], the values computed are close to those highlighted by a previous study focusing on a different animal model (i.e., porcine), which evaluated the same articulation with the same testing method [29].

Regarding the impact of OA progression on AC mechanical properties, a decrease in tissue viscoelastic behavior has been highlighted through OA [27,39]. In this study, AC E and τ decreased from 3 to 6 months, in agreement with OA progression and with the literature [27,40]. For this perspective, while firmly remembering that the main purpose of this study was not to argue about the efficacy of the investigated treatments but to examine the correlation and multicollinearity between the investigated variables, SVF showed a promising ability to limit the reduction in the viscoelastic parameters of AC through the progression of early OA, supporting previous findings about biological and histomorphological features, as extensively reported in [17,18].

Prior to discussing the peculiarities of the proposed ML approaches, and considering that, according to the best of the authors’ knowledge, no study has investigated possible relationships between AC mechanical properties and knee inflammatory condition, the following paragraphs offer insights into the eligibility of the retrieved relationships.

Although OA had been considered to be a prototypical, non-inflammatory disease compared to rheumatoid arthritis (RA) [40], more recent studies highlighted that levels of synovitis may also be high in patients affected by OA [41,42]. IL-6 has been implicated in OA cartilage degeneration and its level correlates with disease incidence and severity [43]. Nevertheless, IL-6 has both catabolic and protective effects in AC, suggesting that its role in OA is not yet completely understood. From this perspective, studies focusing on IL-6 signaling have suggested its key role is in regulating matrix synthesis and degeneration, which in turn affect the cartilage mechanical response through degradation and OA [28].

As in the case of IL-6, PGE2 has been shown to be physiologically relevant for maintaining AC homeostasis [44]. Increasing evidence has defined PGE2 signaling as a potential pathway to protect tissues from inflammation, especially considering its distinct and sometimes opposing role in defining proteoglycan accumulation/synthesis and the collagen II/I (COLL II/I) ratio [45]. Moreover, selective stimulation of the PGE2 signal promoted regeneration of AC tissues with a physiological osteochondral boundary, which is important to maintaining the mechanical and biological properties of AC [46].

Both of these inflammatory factors clearly showed some relationship with the viscoelastic response of AC. Considering their role in regulating cartilage matrix synthesis and degradation, the relationship with AC mechanical behavior is an important finding of this investigation, which was due to the machine learning approach used.

ML approaches have been previously applied to investigating and predicting both the AC mechanical properties [47] and OA progression [48]; however, to the best of the authors’ knowledge, no study has tailored these methodologies and developed regression models to predict AC functionality starting from knee joint inflammation. Herein, VSURF and XGBoost ML algorithms were used for regression analysis. Although these algorithms serve similar purposes, their approaches to handling multicollinearity differ. VSURF indirectly helps in dealing with multicollinearity by selecting variables that contribute unique information to the model. If a set of variables is highly correlated, VSURF may select only one of them, effectively reducing multicollinearity in the final model. XGBoost can handle multicollinearity to some extent by assigning lower feature importance to highly correlated variables and reducing their impact on the model. Additionally, the regularization techniques used in XGBoost, such as shrinkage and column subsampling, help in controlling the influence of correlated variables and improving the model’s robustness.

Regarding the use of various metrics in estimating the accuracy of the regression models adopted here, they allow a more comprehensive evaluation of the model’s performance, in addition to providing different perspectives of their predictive accuracy and enabling informed decisions about the model’s reliability and suitability for a given problem. R^2^ gives an indication of how well the model explains the variance, while RMSE, rRMSE, MAE, and SDe provide information about the magnitude and distribution of the prediction errors.

Although ML algorithms have been previously used with small datasets [49,50], using VSURF and XGBoost on a small dataset might present certain risks and challenges that should be considered. Both VSURF and XGBoost can capture complex relationships in the data but, with small datasets, they may find patterns that are specific to the training data, leading to poor performance on new data. With small datasets, model performance can be very sensitive to specific instances of the set; small variations in the data can have a significant impact on model behavior and results. In addition, there can be a higher ratio of noise (i.e., outliers) to signal. Noise can mislead algorithms and affect the accuracy of variable selection or model fitting, leading to less reliable results.

Prior to summarizing the obtained evidence, a few caveats must be highlighted. Considering both the absence of related literature and the pilot implementation of the proposed approach, the meaning behind the relationships between AC functionality and synovial fluid inflammation is only suggested, specifically by presenting possible connections. Moreover, and still relating to the previous point, a broad investigation should be carried out prior to generalizing our findings; from this perspective, a starting point could be studies based on small animal OA models, due to their availability and cost [51].

Despite the above-mentioned limitations, the retrieved evidence should be taken into account with the aim of developing a multidisciplinary framework to comprehensively study OA features. From this perspective, obtaining information about AC features starting from synovial fluid analysis could be a game-changing approach in understanding the onset and progression of OA, especially considering the potential implications of obtaining complete knowledge about the condition of the knee articular tissues from a diagnostic tool that can be employed from the onset of the early stages of degenerative pathologies.

## 4. Materials and Methods

The was an in vivo, prospective, interventional study in which mild to moderate OA was induced in 24 female Bergamasca_Masseses sheep (47 ± 5 Kg), as reported in Appendix A (see paragraph Study Design) and as indicated in the previously published articles [17,18]. The treatments were performed according to Table 2.

Macroscopic score, histology, histomorphometry, and immunohistochemistry were used to confirm OA in the study groups, and the results were detailed in previous studies [17,18].

### 4.1. Biochemistry of Synovial Fluid

Inflammatory factors of the synovial fluid were quantified with ELISA kits, according to manufacturer instructions (ABclonal Technology, Woburn, MA, USA): interleukin 1β (IL1β) (minimum detectable dose (MDD) typically less than 3.9 pg/mL); interleukin 6 (IL6) (MDD typically less than 0.7 pg/mL); tumor necrosis factor α (TNFα) (MDD typically less than 1.82 pg/mL); cross-linked C Telopeptide of Type II Collagen (CTx2) (MDD typically less than 39 pg/mL); and prostaglandin E2 (PGE2) (MDD typically less than 10 pg/mL). All ELISA kits have an intra-assay CV < 10% and inter-assay CV < 15%.

### 4.2. Gross Evaluation of Knee Joint

A macroscopic score (Gross Articular Damage Score—DS) was determined on lateral and medial tibial plateau and femoral condyles (four quadrants) (the details are in Appendix A) [17,21].

Then, before decalcification to perform histological and histomorphometric analyses, each lateral femoral condyle was divided into two halves; one half was frozen at −20 °C for the subsequent mechanical analyses, and the other half was processed for histology and histomorphometry.

The lateral compartment of the femoral condyles was evaluated because it is the most affected in the lateral meniscectomy animal model, as also documented in the literature [19].

### 4.3. Histology, Immunohistochemistry, and Histomorphometry of Cartilage

The halves of the lateral condyles dedicated to histology were decalcified in formic acid/hydrochloric acid for about 1 month and then processed for paraffin embedding as indicated in a previous study [17]. Three slides of the central portion of the condyles were stained with Safranin O/Fast Green staining (Sigma-Aldrich, St. Louis, MO, USA) and the other three were immunostained for COLL II, using primary antibodies (anti-COLL II). Each slide was acquired with a Aperio ScanScope digital scanner (Aperio ScanScope CS, Aperio Technologies, Leica Biosystems, Milan, Italy). The three slides stained with Safranin O/Fast Green were histomorphometrically evaluated by measuring cartilage thickness (CT, µm) and fibrillation index (FI, %). In the three slides immunostained for COLL II content (%), the ratio between immunopositive stained areas and the total region of interest (ROI) areas was calculated.

### 4.4. Mechanical Analysis of Cartilage

The halves of the lateral condyles dedicated to mechanical evaluation were tested as fresh, i.e., by applying indentation mapping technique to the AC [19]. On the day of the test, specimens were thawed at room temperature and allowed to equilibrate for 1 h in saline solution (0.9% NaCl, Baxter S.P.A) [22]. Subsequently, indentation mapping of the entire specimen surface was performed following the protocol described in [18]. Cartilage thickness (TH, µm) in the previously indented positions was experimentally achieved via a needle technique, using a 27G 1/2” Precision Glide intradermal bevel needle (BD, Franklin Lakes, NJ, USA) [52]. Then, the cartilage Young’s modulus (E, MPa) was computed by fitting the indentation load–displacement ramp data to the Hayes model [53] with TH as the fixed parameter, while relaxation time (τ, s) was obtained by considering the time taken by the tissue to reach an equilibrium load value during constant indentation displacement [53]. Each indentation map was averaged to obtain a single (E, τ) for each specimen.

Finally, the relationship among computed mechanical parameters (dependent variables: E and τ) and previously investigated macroscopic and histomorphometric parameters (DS, CT, FI, and COLL II), synovial soluble factors (IL1β, IL6, TNFα, CTx2, and PGE2), and treatment type (GROUP) and experimental time (TIME) as potential predictors, were investigated. A diagram of the experimental workflow is depicted in Figure 3.

### 4.5. Statistical Analysis

The statistical analysis was carried out using the software R v. 4.2.1 [54] and various R packages for implementation of machine learning models (VSURF v. 1.1.0 [55] and ‘xgboost’ [56]) and ‘ggplot2’ v.3.1.1 [54]). Data are reported as mean ± SD at a significance level of *p* < 0.05.

After verifying normal distribution (Shapiro–Wilks test) and variance homogeneity (Levene test) of data, two-way ANOVA was used to test if significant effects or interactions of factors on measured mechanical parameters were present; the type of treatment administered (“Treatment”, 4 levels: NaCl, ADSCs, AECs, SVF) and the experimental time (“Time”, 2 levels: 3 and 6 months) were considered as fixed factors. Pairwise comparisons of estimated marginal means were carried out as post hoc tests to identify significant differences among groups. Sidak’s adjusted *p*-values were calculated.

To evaluate the relationship among mechanical parameters and previously measured histomorphometric and synovial fluid parameters [57], a preliminary correlation analysis was performed using Spearman’s rank correlation coefficient. Then, to reduce multicollinearity in regression models, machine learning (ML) algorithms—Variable Selection Using Random Forests (VSURF) and Extreme Gradient Boosting (XGB)—were used to firstly rank the variables according to their variable importance from a trained predictive model, eliminating those of small importance according to an estimated threshold, and secondly select remaining variables for interpretation and predictions [58]. In addition, linear models considering multicollinearity were used. A K-fold cross-validation technique was used to assess the model’s performance and effectively tune XGB hyperparameters. The estimation and effectiveness of all selected regression models was evaluated by calculating R^2^, root mean squared error (RMSE), relative RMSE (rRMSE), mean absolute error (MAE), and standard deviation of estimation (SDe) [58]. More details on these estimators are reported in Appendix A.

## 5. Conclusions

When aiming to properly assess the impact on joints of extremely complex pathologies such as OA, a comprehensive evaluation of the joints’ main features should be considered. The present study represents a step forward in relating AC functionality to the inflammatory condition of the knee joint, based on ML approaches to build on information retrieved from a prospective, interventional study of a large OA animal model. The treatment of such a model using orthobiological solutions, i.e., ADSCs, SVF, and AECs, represents a promising context for identifying relationships between articular tissues impaired by OA. The ML algorithms highlighted that AC viscoelastic parameters, i.e., lnE and τ, can be predicted starting from inflammatory factors of synovial fluid, i.e., IL6 and PGE2, and tissue histological features, i.e., CT. In addition, SVF confirmed its efficacy, in comparison to the other treatments, in limiting the reduction in the AC viscoelastic parameters through the progression of early OA, supporting the findings reported in our previous studies.

The relationships revealed herein suggest the possibility of retrieving information about the functional features of AC—and how OA impairs them—starting from the results of a diagnostic tool, i.e., synovial fluid analysis. This evidence supports the evaluation of OA as a whole-joint disease, in addition to providing crucial insights about potential biological targets for AC regenerative therapies. Therefore, future studies should closely investigate the mechanisms behind relationships between AC functionality and joint inflammation conditions, thereby improving the knowledge about the onset of degenerative pathologies and, most importantly, properly developing relative treatments.

## Figures and Tables

**Figure 1 ijms-24-13374-f001:**
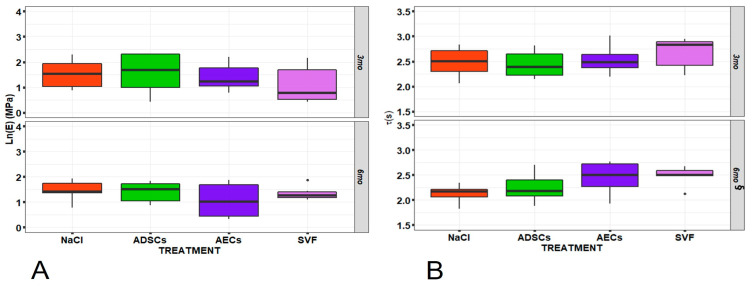
Boxplots of (**A**) Young’s modulus (lnE) and (**B**) relaxation time (τ) for each treatment, i.e., adipose-derived mesenchymal stem cells (ADSCs), stromal vascular fraction (SVF), amniotic endothelial cells (AECs), and control (NaCl), and experimental time (3 and 6 months). Two-way ANOVA did not show any significant interactions or effects for lnE, while a significant effect was found for τ (time: *F* = 7.01, *p* = 0.011). Pairwise comparisons (1 symbol: *p* < 0.05; 2 symbols: *p* < 0.005; 3 symbols: *p* < 0.0005): ^§^, 6 months versus 3 months. The central mark indicates the median, the bottom and top edges of the box indicate the 25th and 75th percentiles, respectively, whereas whiskers represent minimum and maximum valu Black dot = An outlier that is an observation that lies an abnormal distance from other values in a random sample from a population.

**Figure 2 ijms-24-13374-f002:**
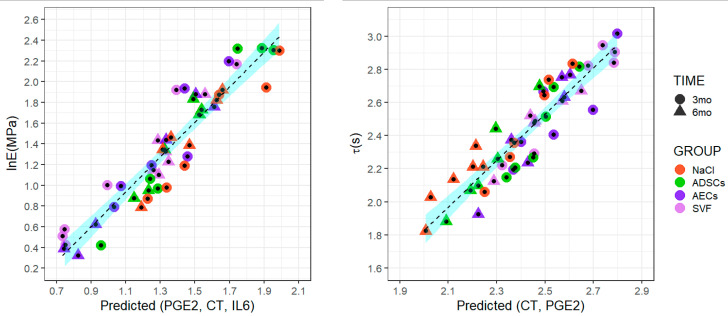
Scatter plots of elastic modulus (lnE) and relaxation time (τ) predicted versus actual values, stratified for treatment (saline solution—NaCl, adipose-derived mesenchymal stem cells—ADSCs, amniotic epithelial stem cells—AECs, stromal vascular fraction—SVF) and experimental time (3 and 6 months). The line and confidence bounds are computed using the locally estimated scatterplot smoothing (loess).

**Figure 3 ijms-24-13374-f003:**
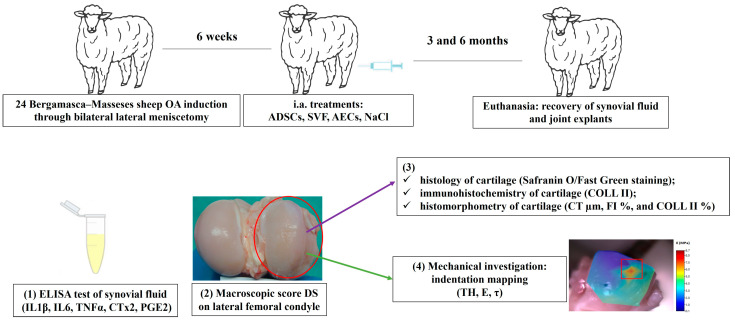
Scheme of in vivo experimental set up and the performed analysis: (1) Biochemistry of synovial fluid; (2) Gross evaluation of knee joint; (3) Histology, immunohistochemistry, and histomorphometry of cartilage; (4) Mechanical analysis of cartilage. OA = osteoarthritis; ADSCs = adipose-derived mesenchymal stem cells; SVF = stromal vascular fraction; AECs = amniotic endothelial mesenchymal cells; NaCl = saline solution; IL1β = interleukin 1 beta; IL6 = interleukin 6; TNFα = tumor necrosis factor alpha; CTx2 = cross-linked C Telopeptide of Type II Collagen; PGE2 = prostaglandin E2; DS = damage score; COLL II = Collagen II; CT = cartilage thickness; FI = fibrillation index; TH = cartilage thickness; E = Young’s modulus; τ = relaxation time. Red circle and square are the damage area of cartilage on which analyses were performed.

**Table 1 ijms-24-13374-t001:** The estimation and effectiveness of the final models (model performance and validation) expressed as R^2^, root mean squared error (RMSE), mean absolute error (MAE), and standard deviation of estimation (SDe) for MLR, VSURF, and XGBR models with selected predictors.

Parameter	Methods	Selected Predictors	R^2^	RMSE	rRMSE	MAE	SDe	ANOVA among Abs (Residuals)
lnE	LR	CT *	0.002	0.55	36.6	0.42	0.76	*F* = 3.16, *p* = 0.058
	VSURF	PGE2, CT, IL6	0.87	0.30	21.8	0.25	0.07	
	XGBR	PGE2, IL6, CT	0.012	0.81	53.2	0.68	1.94	
τ	LR	CT °	0.52	0.15	6.4	0.14	0.09	*F* = 2.99, *p* = 0.067
	VSURF	CT, PGE2	0.84	0.14	5.9	0.12	0.01	
	XGBR	IL6, CT, PGE2	0.35	0.22	8.5	0.18	0.10	

* InE: *F* = 3.55, *p* = 0.068; ° τ: *F* = 10.23, *p* = 0.003. However, considering the model that determined the lowest MAE values, i.e., that for τ; the higher R^2^, the better; and the smaller RMSE and rRMSE, the better, the results more in line with these considerations were those obtained with the ML algorithm of VSURF.

**Table 2 ijms-24-13374-t002:** Treatment schemes.

Treatment	Processing	Joints (n)	Experimental Time (Months)
Stromal Vscular Fraction-SVFs (1 mL)	After 6 weeks from meniscectomy (on the day of treatment), animals underwent to inguinal adipose tissue removal under general anaesthesia. It was digested with 0.075% collagenase II for 1 h and then the reaction was stopped with complete medium. Cells were centrifuged injected.	6	3
		6	6
Amniotic Endothelial Cells—AECs (2.5 × 10^6^ cells/mL, 1 mL)	On the day of the treatment, these cells were obtained from “Unit of Basic and Applied Biosciences” of Bioscienze e Tecnologie Agro-alimentari e Ambientali Faculty, University of Teramo, Teramo (Italy). AECs were previously in vitro expanded for 3 passages in an Eagle-α modification medium, supplemented with 20% FBS, 1% ultraglutamine and 1% penicillin/streptomycin without any growth factors, before delivering them to the surgery room.	6	3
		6	6
Adipose-Derived Mesenchymal Stem Cells—ADSCs (2.5 × 10^6^ cells/mL, 1 mL)	On the day of meniscectomy, abdominal adipose tissue was harvested and ADSCs were obtained after in vitro expansion. It was digested with 0.075% collagenase II and the enzymatic reaction was stopped by the addition of DMEM supplemented with 10% FBS, 100 U/mL penicillin, 100 mg/mL streptomycin and 5 mg/mL plasmocin. Cells were centrifuged and the nucleated ones were then seeded in complete medium. At subconfluence at P2, the adherent cells were detached and injected 6 weeks later.	6	3
		6	6
0.9% Sodium Chloride—NaCl (1 mL)	Sterile saline solution.	6	3
		6	6

## Data Availability

The raw data supporting the conclusions of this article will be made available by the authors, upon reasonable request, without undue reservation.

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
