# Peer review of "Relations between Structure/Composition and Mechanics in Osteoarthritic Regenerated Articular Tissue: A Machine Learning Approach"

_ijms, 2023, doi:10.3390/ijms241713374_

Round 1

Reviewer 1 Report

The authors treated osteoarthritis in sheep, with three different preparations of mesenchymal stem cells, however, in the methods section it is not mentioned, how osteoarthritis was induced, and what was the degree of osteoarthritis. In Figure one, it can be read that the method was bilateral lateral meniscectomy. Was it lateral or bilateral? The results of the treatment are also not clear. The therapy is not adequately described, how many cells were injected? how were the cells prepared, and how was the expansion performed? In Figure two, it is not clear what is shown, was the treatment benefit the animals or not? what treatment was better? The discussion is not focused and obscure, and the conclusions do not state if the therapy was successful or not. In conclusion, it is said that "... a context for searching which tissues can affect AC functions..." but the tissues are not named. So a question mark remains also in the conclusions: Which tissues affect AC? Finally,  what has this study found?

Author Response

We want to thank the reviewer for her/his thorough review of the article and her/his suggestions for improving the manuscript. Below are the changes, with line and page numbers, while in the text the changes are shown in red.

Many thanks from all the authors.

The authors treated osteoarthritis in sheep, with three different preparations of mesenchymal stem cells, however, in the methods section it is not mentioned, how osteoarthritis was induced, and what was the degree of osteoarthritis.

As reported in the “Introduction” section, and briefly described in the “Materials and Methods” section, information regarding the development of the animal model for OA and the results obtained from treatment with the three different MSC preparations have been reported in previously published articles of our group:

  1. Veronesi, F.; Fini, M.; Martini, L.; Berardinelli, P.; Russo, V.; Filardo, G.; Di Matteo, B.; Marcacci, M.; Kon, E. In Vivo Model of Osteoarthritis to Compare Allogenic Amniotic Epithelial Stem Cells and Autologous Adipose Derived Cells. Biology (Basel) 2022, 11, 681.
  2. Veronesi, F.; Berni, M.; Marchiori, G.; Cassiolas, G.; Muttini, A.; Barboni, B.; Martini, L.; Fini, M.; Lopomo, N.F.; Marcacci, M.; Kon, E. Evaluation of cartilage biomechanics and knee joint microenvironment after different cell-based treatments in a sheep model of early osteoarthritis. Int Orthop 2021, 45, 427-435.

In order not to create repetitions, we have added the following brief sentence in the text:“The in vivo study, a prospective, interventional study, carried out in 24 female Bergamasca_Masseses sheep (47 ± 5 Kg), was performed to induce mild to moderate OA, as reported in Appendix A (see paragraph Study Design) and as indicated in the previously published articles [17, 18].” (“Materials and methods” section, page 7, lines 275-278).

Briefly, at T0 in open surgery under general anesthesia, in both legs, the cranial and lateral attachments of the lateral meniscus were transected, removing completely the meniscus. The cartilage was not damaged. All surgeries were performed by the same veterinary surgeon [17].” (“Appendix A” section, page 13, lines 477-480).

The degree of OA was “mild to moderate OA” as indicated in the text: “…to induce mild to moderate OA….” (“Materials and methods” section, page 7, line 276).

In Figure one, it can be read that the method was bilateral lateral meniscectomy. Was it lateral or bilateral? The results of the treatment are also not clear.

The term could be misleading, but with "Bilateral lateral meniscectomy" we mean that the meniscectomy is lateral (it occurs only in the “lateral” meniscus), but it is performed in both joints ("bilateral"). This is a term that we have used both in drafting the project and in previous published articles, mentioned in the text.

In our previous two studies [references 17 and 18], we have the results of the treatments concluding that all the three treatments showed better results than control (injection of NaCl), but SVF and AECs showed superiority over ADSCs, because they induced higher cartilage regeneration and lower inflammation. SVF showed better results than AECs at 3 and 6 months. So, SVF seems to be more favorable than the other biological options, because it is easily obtained and rapidly used after harvesting, with good healing potential. AECs cause no discomfort and could be also considered for the treatment of OA joints.

However, the aim of the study was to reveal existing relations between articular tissue structure/composition – investigated by biochemistry and histology (previously reported [17, 18] – and mechanics – assessed by indentation – in the case of various orthobiological treatments responding differently to the progression of such a disease.

The therapy is not adequately described, how many cells were injected? how were the cells prepared, and how was the expansion performed?

Aiming to better clarify these aspects, Table 1, now Table 2, has been improved, i.e., by adding a summary of the preparation methodology and quantity of injected cells.

In Figure two, it is not clear what is shown, was the treatment benefit the animals or not? what treatment was better?

As reported in Figure 2, now Figure 1, and, as stated in the “Results” section, there are no significant results testifying the impact of the applied cell treatments on the articular cartilage (AC) mechanical response. Only a modest trend of higher viscous parameter is observed in cases treated with SVF at 6 months. In this perspective, previous studies highlighted a decrease of the viscous response – i.e., relaxation time, τ – of AC through OA [37, 39, 46], supporting the finding reported by this study. This information is also added to the Discussion: “Focusing on the extent of the mechanical properties of AC here reported, Young’s modulus values agree with previous studies concerning the same animal model [36, 37]. Regarding τ, no study investigated its extent in the same animal model and applying the same testing method. Despite τ can be highly variable and dependent on site, articulation, model from which AC is retrieved, and testing method [27, 38], the values computed are close to the ones highlighted by a previous study focusing on a different animal model – i.e., porcine –, which evaluated the same articulation with the same testing method [39].(“Discussion” section, page 5, lines 191-197).

The discussion is not focused and obscure, and the conclusions do not state if the therapy was successful or not. In conclusion, it is said that "... a context for searching which tissues can affect AC functions..." but the tissues are not named. So, a question mark remains also in the conclusions: Which tissues affect AC? Finally, what has this study found?

According to reviewer suggestions, “Discussion” and “Conclusions” sections have been now revised (in red), aiming to clarify better these aspects.

Reviewer 2 Report

My biggest doubt in the presentation of materials, methods, and results concerns the methodology of confirming or excluding the development of surgically induced OA. What features (clinical signs, radiological signs, histopathological features) were used to confirm OA in the control group (normal saline) and in the three study groups (ADSCs, AECs, and SVFs)? 

Moreover, in the results section, it is advisable to present the measurement values for each considered feature, not only for Young's modulus and relaxation time. If only Young's modulus, relaxation time, and the relationships between features are to be presented in this manuscript, the whole manuscript body needs to be thoroughly rewritten and referenced to other articles created on the basis of the same research model in which other data described in the experiment were presented.

Detailed comments

L 16 Add coma before 'and histomorphometry'.

L 18 Remove 'IL6' before 'Interleukin'.

L 18-19 Interleukin and Prostaglandin should be lowercase

L 19-20 There was no data on Stromal Vascular Fraction evaluation in the above part of the abstract section, please provide it.

L 20 Expand the abbreviation when using it for the first time.

L 26 And animal joint. It is important since you use a large animal model of OA.

L 34 And 35 What do you mean by 'therapeutic infiltrative strategy' and 'infiltrative conservative treatments'?

L 37 What about anti-inflammatory treatment, rehabilitation, and cartilage supplementation in the early stage of OA? 

L 43 Please, provide an appropriate reference.

L 49 As 'many preclinical and clinical studies' are reported, adequate basic references should be provided.

L 50 Remove the semicolon and start a new sentence. 

L 52-57 It would also be advisable to refer to previous studies other than our own, using a large animal model, especially a sheep model, in the study of OA. There are many such studies. Moreover, a few sentences of introduction why sheep are considered the very best, if not the best, large animal model of OA in translational research is indicated.

L 58 Are you sure you are researching 'the impact of OA on the homeostasis of AC'? Consider whether it would not be better to specify this introductory sentence so that it is not confusing.

L 'over-stresses' Please, provide an appropriate reference.

L 64 The total collagen or the specific type?

In the introduction section, the justification for investigating IL1β, IL6, TNFα, CTx2, and PGE2 as well as DS, COLL II, CT, FI, TH, E, and τ should be provided in the link of known OA pathogenesis.

L 77 Essential pieces of information that enable the reader to assess the quality of the study are not provided. These include for example type of the study design (prospective/retrospective? observational/interventional?), the use of blinding/randomization, and the calculations for sample size determination (was a power analysis performed?). The most critical population data of selected sheep (age, gender, body weight) should also be added.

L 80 Remove the extra dot in the table header. Also, expand all abbreviations used in the table, as each table should be self-readable regardless of the contents of the manuscript.

The treatment protocol should be described in detail. The brief description contained in Table 1 is for reference only and does not allow for the experiment to be repeated by other researchers. Was in any case day 0 of the meniscectomy. How was meniscectomy performed by arthrotomy or arthroscopy? Was the procedure performed under general anesthesia? Whether from the lateral or medial approach to the knee joint? Were both knees operated on each sheep? Was the selection of knee joints randomized? What anti-inflammatory drugs, or antibiotics did the sheep receive after the procedure? Or all sheep? How long? Was each intra-articular infusion (ADSCs, AECs, SVF, or control saline) performed 6 weeks after meniscectomy? Was it performed under general or local anesthesia? Were anti-inflammatory drugs and/or antibiotics given after the intra-articular infusion? How was the intra-articular access site secured after the intervention? How were the animals kept, individually or in groups? How were the sheep euthanized 3 or 6 weeks after administration of ADSCs, AECs, SVF, or control saline? How were the samples taken? All this information is necessary for the reader to properly understand the presented research results. I agree, you can use references to describe the preparation of ADSCs, AECs, and SVFs, but the description of the animal experiment should be complete.

L 82 Were these commercial tests? Was the test specific for the sheep specimen? Please, provide sensitivity, intraassay CV, and interassay CV for each ELISA kit. What sample dilution was used? Was a biochemical, cytological, and bacteriological examination of the synovial fluid performed?

L 86 The table containing the DS scoring system should be provided.

L 87 Were X-rays or CT scans of the knee joints taken to assess radiological signs of OA? Were such images taken prior to the procedure to rule out endogenous OA? Were goniometric measurements of the joint performed?

L 95 was primary H&E staining done?

L 100 Expand the abbreviation (COLL II) when using it for the first time.

L 102 Remove the additional dot.

L 127 Figure 1. In the material and method section, there is nothing about H&E staining and classification of the histopathological results. All abbreviations used should be expanded in the caption of Figure 1.

L 156 In the results section, it is advisable to present the measurement values for each considered feature, not only for Young's modulus and relaxation time. Otherwise, describing their measurements in the materials and methods section is not justified.

L 171 Figure 2. All abbreviations used should be expanded in the caption of Figure 2 as well as the data shown in the plots. Do boxes represent the lower quartile, median, and upper quartile, whereas whiskers represent minimum and maximum values? Why are you using boxing plots since you declared in L 135 that data have verified normal distribution?

L 174 There are no 1 symbol: p<0.05; 2 symbols: p<0.005; 3 symbols: p<0.0005 on a plot.

The discussion also requires to be restructured. In its current form, no apparent logical order is used in comparing/contrasting the results obtained in the study with previous literature. The authors should attempt structuring the discussion following a "common standard format" that usually consists of the following points:

1. One-sentence summary that highlights the most relevant results.

2. A thorough discussion of each result obtained in relation to the corresponding study objective: was the tested hypothesis confirmed or not? Why? What previous evidence supports the specific result or not? It is critical to compare/contrast the result obtained with previous literature on the sheep first, then in other large animal models, and finally in human medicine.

c. Statement of study limitations

d. Future directions

e. Conclusions

The conclusion section requires a complete rewrite so that it indicates the conclusions of the presented results and not general conclusions regarding the use of selected techniques in the large animal model of OA.

After implementing all the suggested changes, the results obtained might significantly differ from those currently reported in the manuscript. Thus, no specific comments are provided below, and a more thorough review of the discussion section will be provided once the manuscript is considered overall scientifically sound.

Author Response

We want to thank the reviewer for her/his thorough review of the article and her/his suggestions for improving the manuscript. Below are the changes, with line and page numbers, while in the text the changes are shown in red.

Many thanks from all the authors.

My biggest doubt in the presentation of materials, methods, and results concerns the methodology of confirming or excluding the development of surgically induced OA. What features (clinical signs, radiological signs, histopathological features) were used to confirm OA in the control group (normal saline) and in the three study groups (ADSCs, AECs, and SVFs)?

For the sake of brevity, since in this article only the biomechanical results are shown, while the biological and histological ones have already been published [Ref n° 17 and 18], we thought only of mentioning this argument, and reporting everything to the two articles already published. In fact, this present article is a corollary of our two previous ones. However, as requested by the reviewer, details about the assessment of the surgically induced OA have been reported in “Materials and methods” section. In particular, damage of the articular surfaces of each knee was assessed by macroscopic score, histology, immunohistochemistry and histomorphometry. So, a new sentence has now been added as follows “Macroscopic score, histology, histomorphometry and immunohistochemistry were used to confirm OA in the study groups, and the results were detailed in previous studies [17, 18].” (“Materials and methods” section, page 7, lines 279-281).

Moreover, in the results section, it is advisable to present the measurement values for each considered feature, not only for Young's modulus and relaxation time. If only Young's modulus, relaxation time, and the relationships between features are to be presented in this manuscript, the whole manuscript body needs to be thoroughly rewritten and referenced to other articles created on the basis of the same research model in which other data described in the experiment were presented.

The purpose of the present study is to reveal possible relationships between articular structure/composition and the mechanical response of AC, i.e., by suggesting the use of a machine learning approach to develop a model capable of predicting – by a combination of various histological and biochemical factors– changes of the viscoelastic properties of the AC through OA. Details about biochemistry of synovial fluid, gross evaluation of knee joint, histology immunohistochemistry and histomorphometry of AC – along with the results of such analyses – have been reported in a previously published article [17, 18]. In this perspective, in the present study only the mechanical evidence and the machine learning approaches are fully reported and highlighted. Despite the manuscript is revised according to the reviewers’ suggestions, it is noteworthy to mention that, based on the authors knowledge, no study has proposed – to date – an approach similar to the one here reported, i.e., by using machine learning to predict the mechanical properties of AC starting by the biological-histomorphological features.

Detailed comments:

L 16 Add coma before 'and histomorphometry'. Done.

L 18 Remove 'IL6' before 'Interleukin'. Done.

L 18-19 Interleukin and Prostaglandin should be lowercase. Done.

L 19-20 There was no data on Stromal Vascular Fraction evaluation in the above part of the abstract section, please provide it. Done.

L 20 Expand the abbreviation when using it for the first time. Done.

L 26 And animal joint. It is important since you use a large animal model of OA. Done.

 L 34 And 35 What do you mean by 'therapeutic infiltrative strategy' and 'infiltrative conservative treatments'?

Therapeutic infiltrative strategy’ means strategies that use the injection of therapeutic substances – usually painkillers, local anesthetics and anti-inflammatory agents – directly into a joint. In this perspective, 'infiltrative conservative treatments’ refers to the substances – injected at the intra-articular level – offering a rapid return to daily activities due to their ability of reducing pain and inflammation, without the involvement of invasive surgical procedures. The ‘infiltrative’ and ‘conservative’ terms are often used without distinction.

L 37 What about anti-inflammatory treatment, rehabilitation, and cartilage supplementation in the early stage of OA?

Recently, a clinical practice guideline has been drawn up to summarize the management of knee OA through nonarthroplastic methods. The strong recommended therapies are the use of lateral wedge insoles, supervised exercise, unsupervised exercise, and/or aquatic exercise, education programs, topical or oral NSAIDs, and oral acetaminophen or narcotics. So, a new sentence has been added to the “Introduction” as follows: “Among the most recommended non-surgical treatments we mention the use of lateral wedge insoles, supervised exercise, unsupervised exercise, and/or aquatic exercise, education programs, topical or oral NSAIDs, and oral acetaminophen or narcotics [5].” (“Introduction” section, page 1, lines 40-43).

In addition, the following reference has been added: “5. Brophy, R.H.; Fillingham, Y.A. AAOS Clinical Practice Guideline Summary: Management of Osteoarthritis of the Knee (Non-arthroplasty), Third Edition. J Am Acad Orthop Surg 2022, 30, e721-e729.” (“References” section, page 15, lines 542-543).

L 43 Please, provide an appropriate reference.

A new reference (n° 9) has now been added in the “Introduction” and “References” sections: “8. Carneiro, D.C.; Araújo, L.T.; Santos, G.C.; Damasceno, P.K.F.; Vieira, J.L.; Santos, R.R.D.; Barbosa, J.D.V.; Soares, M.B.P. Clinical Trials with Mesenchymal Stem Cell Therapies for Osteoarthritis: Challenges in the Regeneration of Articular Cartilage. Int J Mol Sci 2023, 24, 9939.” (“References” section, page 15, lines 549-551).

L 49 As 'many preclinical and clinical studies' are reported, adequate basic references should be provided.

In order to add not too many references, we had decided to include only this reference as it is a systematic review that reports evidence on the use of mesenchymal cells in studies on cartilage regeneration.

L 50 Remove the semicolon and start a new sentence. Done.

L 52-57 It would also be advisable to refer to previous studies other than our own, using a large animal model, especially a sheep model, in the study of OA. There are many such studies. Moreover, a few sentences of introduction why sheep are considered the very best, if not the best, large animal model of OA in translational research is indicated.

As suggested by the reviewer, the following sentence has been added to the “Introduction” section: “In pre-clinical studies focusing on OA, large animal models are usually preferred, mainly because i) the anatomy of their joints is more similar, compared to small animal models, to that of humans, and ii) due the possibility to extract and analyze synovial fluid [14]. In this perspective, and taking into consideration anatomy, size and nature, as well as relatively lower associated costs and rising popularity, sheep represents an excellent model for such studies [15]. Considering the suitability of MSCs in the treatment of OA, few studies investigated their potential efficacy, mainly – but not only, focusing on in vivo large animal models, as sheep [16-18].” (“Introduction” section, page 2, lines 60-68).

Consequently, new references have been added to the “References” section: “14. Chu, C.R.; Szczodry, M.; Bruno, S. Animal models for cartilage regeneration and repair. Tissue Eng Part B Rev 2010, 16, 105-115.

  1. Music, E.; Futrega, K.; Doran, M.R. Sheep as a model for evaluating mesenchymal stem/stromal cell (MSC)-based chondral defect repair. Osteoarthritis Cartilage 2018, 26, 730-740.
  2. Lv, X.; He, J.; Zhang, X.; Luo, X.; He, N.; Sun, Z.; Xia, H.; Liu, V.; Zhang, L.; Lin, X.; Lin, L.; Yin, H.; Jiang, D.; Cao, W.; Wang, R.; Zhou, G.; Wang, W. Comparative Efficacy of Autologous Stromal Vascular Fraction and Autologous Adipose-Derived Mesenchymal Stem Cells Combined With Hyaluronic Acid for the Treatment of Sheep Osteoarthritis. Cell Transplant 2018, 27, 1111-1125.” (“references” section, page 15, lines 557-563).

L 58 Are you sure you are researching 'the impact of OA on the homeostasis of AC'? Consider whether it would not be better to specify this introductory sentence so that it is not confusing. L 'over-stresses' Please, provide an appropriate reference.

The sentence about the homeostasis refers to the impact of OA on the main features of AC considering a more general point-of-view. It is well known that, in joint diseases as OA, AC homeostasis is disrupted by mechanisms that are driven by combinations of biological mediators that vary according to the disease process, including contributions from other joint tissues [20]. Considering the relative literature, we believe that the sentence’s meaning is correct.

A reference has been added supporting over-stresses as a cause of OA: 19. Buckwalter, J.A.; Anderson, D.D.; Brown, T.D.; Tochigi, Y.; Martin, J.A. The Roles of Mechanical Stresses in the Pathogenesis of Osteoarthritis: Implications for Treatment of Joint Injuries. Cartilage 2013, 4, 286-294. (“references” section, page 15, lines 573-574).

L 64 The total collagen or the specific type?

In the introduction section, the justification for investigating IL1β, IL6, TNFα, CTx2, and PGE2 as well as DS, COLL II, CT, FI, TH, E, and τ should be provided in the link of known OA pathogenesis.

Despite the reference study was focused to the total collagen amount, it is reasonable to suppose that it was referred to type II collagen which is one of the main components of AC, in healthy conditions. Justification about the features investigated in the study has been added, together with relative references: “Considering the pathogenesis of OA, synovial fluid inflammatory factors – e.g., IL1β, IL6, TNFα, CTx2, and PGE2 [23], tissue composition and appearance – e.g., type II collagen content [24], fibrillation index [25], and tissue thickness [26] – and AC viscoelasticity – e.g., Young’s modulus, E, and relaxation time, τ [27-29] – are just some of the features deregulated and impaired by the onset and progression of such a disease.” (“Introduction” section, page 2, lines 83-87).

23. Sanchez-Lopez, E.; Coras, R.; Torres, A.; Lane, N.E.; Guma, M. Synovial inflammation in osteoarthritis progression. Nat Rev Rheumatol 2022, 18, 258-275.

  1. Poole, A.R.; Kobayashi, M.; Yasuda, T.; Laverty, S.; Mwale, F.; Kojima, T.; Sakai, T.; Wahl, C.; El-Maadawy, S.; Webb, G.; Tchetina, E.; Wu, W. Type II collagen degradation and its regulation in articular cartilage in osteoarthritis. Ann Rheum Dis 2002, 61 Suppl 2(Suppl 2), ii78-81.
  2. Pinamont, W.J.; Yoshioka, N.K.; Young, G.M.; Karuppagounder, V.; Carlson, E.L.; Ahmad, A.; Elbarbary, R.; Kamal, F. Stand-ardized Histomorphometric Evaluation of Osteoarthritis in a Surgical Mouse Model. J Vis Exp 2020, 159, 10.3791/60991.
  3. Wirth, W.; Ladel, C.; Maschek, S.; Wisser, A.; Eckstein, F.; Roemer, F. Quantitative measurement of cartilage morphology in osteoarthritis: current knowledge and future directions. Skeletal Radiol 2022.
  4. Kumar, R.; Pierce, D.M.; Isaksen, V.; Davies, C.L.; Drogset, J.O.; Lilledahl, M.B. Comparison of Compressive Stress-Relaxation Behavior in Osteoarthritic (ICRS Graded) Human Articular Cartilage. Int J Mol Sci 2018, 19, 413.
  5. Nakamura, S.; Ikebuchi, M.; Saeki, S.; Furukawa, D.; Orita, K.; Niimi, N.; Tsukahara, Y.; Nakamura, H. Changes in viscoelastic properties of articular cartilage in early stage of osteoarthritis, as determined by optical coherence tomography-based strain rate tomography. BMC Musculoskelet Disord 2019, 20, 417.
  6. Han, G.; Chowdhury, U.; Eriten, M.; Henak, C.R. Relaxation capacity of cartilage is a critical factor in rate- and integri-ty-dependent fracture. Sci Rep 2021, 11, 9527.” (“References” section, page 15, lines 579-594).

L 77 Essential pieces of information that enable the reader to assess the quality of the study are not provided. These include for example type of the study design (prospective/retrospective? observational/interventional?), the use of blinding/randomization, and the calculations for sample size determination (was a power analysis performed?). The most critical population data of selected sheep (age, gender, body weight) should also be added.

The study was a prospective, interventional one.  It was decided to assign random associations of treatments – i.e., by changing only the implant side (right limb or left limb) –, thus to not involve more than one anaesthesia per animal. More in detail, randomization was performed as follows: i) right: SVF, left: AECs; ii) left: SVF, right: AECs; iii) right: ADSCs, left: 0.9% NaCl; iv) left: ADSCs, right: 0.9% NaCl. Aiming to define properly the main outcomes of the study, a power analysis was performed, which considered i) the different treatments involved – i.e., 4 levels, AECs, ADSCs, SVF and NaCl –, ii) the experimental times – i.e., 2 levels, 3 and 6 months – and iii) that treatments were placed in both knee joints by using the combination reported above. According to these peculiarities, a minimum number of 24 sheep were required, corresponding to n = 6 knee joints for treatment, and experimental time with a power of 80% and a p-value < 0.05. Regarding the selected animal model, female Bergamasca–Masseses sheep (47 ± 5 Kg) was considered.

This information has been added to the manuscript, and in the relative sections of Appendix A as follows: “The in vivo study, a prospective, interventional study, carried out in 24 female Ber-gamasca_Masseses sheep (47 ± 5 Kg), was performed to induce mild to moderate OA, as reported in Appendix A (see paragraph Study Design) and as indicated in the previously published articles [17, 18].” (“Materials and methods” section, page 7, lines 275-278).

 “Aiming to limit the number of anesthesia for each animal, the process of assigning treatments randomly did not interest the whole group of animals. In this perspective, it was decided to assign random associations of treatments – i.e., by changing only the implant side (right limb or left limb) –, thus to not involve more than one anaesthesia per animal. Accordingly, randomization was performed as follows: i) right: SVF, left: AECs; ii) left: SVF, right: AECs; iii) right: ADSCs, left: 0.9% NaCl; iv) left: ADSCs, right: 0.9% NaCl.  With the purpose of arhuing properly about the results, a power analysis was performed, considering i) the different treatments involved – i.e., 4 levels, AECs, ADSCs, SVF and NaCl –, ii) the experimental times – i.e., 2 levels, 3 and 6 months – and iii) that treatments were placed in both knee joints by using the combination reported above. According to these peculiarities, a minimum number of 24 sheep were required, corresponding to n = 6 knee joints for treatment, and experimental time with a power of 80% and a p-value < 0.05.” (“Appendix A” section, page 14, lines 494-506). 

L 80 Remove the extra dot in the table header. Also, expand all abbreviations used in the table, as each table should be self-readable regardless of the contents of the manuscript. Done.

The treatment protocol should be described in detail. The brief description contained in Table 1 is for reference only and does not allow for the experiment to be repeated by other researchers. Was in any case day 0 of the meniscectomy. How was meniscectomy performed by arthrotomy or arthroscopy? Was the procedure performed under general anesthesia? Whether from the lateral or medial approach to the knee joint? Were both knees operated on each sheep?

For the sake of brevity, we have included the bibliographic references of previously published articles on the argument [17, 18]. However, based on the reviewers' suggestions, we have added the following sentences: “Briefly, at T0 in open surgery under general anesthesia, in both legs, the cranial and lateral attachments of the lateral meniscus were transected, removing completely the meniscus. The cartilage was not damaged. All surgeries were performed by the same veterinary sur-geon [17].” (“Appendix A” section, page 13, lines 477-480).

Was the selection of knee joints randomized?

In Appendix A the following sentence has been added: “Aiming to limit the number of anesthesia for each animal, the process of assigning treatments randomly did not interest the whole group of animals. In this perspective, it was decided to assign random associations of treatments – i.e., by changing only the implant side (right limb or left limb) –, thus to not involve more than one anaesthesia per animal. Accordingly, randomization was performed as follows: i) right: SVF, left: AECs; ii) left: SVF, right: AECs; iii) right: ADSCs, left: 0.9% NaCl; iv) left: ADSCs, right: 0.9% NaCl.” (“Appendix A” section, page 14, lines 494-500).

What anti-inflammatory drugs, or antibiotics did the sheep receive after the procedure? Or all sheep? How long?

The following sentence has been added: “For all sheep, the postoperative therapy consisted of cephalosporin 1 g per day for 5 days and analgesics (metamizole sodium 40 mg/kg/die intramuscular, for 3 days; ropivacaine 7.5 mg/mL intramuscular, and Fentanyl 50 µg/72 transdermic patch).” (“Appendix A” section, page 13, lines 480-482).

Was each intra-articular infusion (ADSCs, AECs, SVF, or control saline) performed 6 weeks after meniscectomy? Was it performed under general or local anesthesia? Were anti-inflammatory drugs and/or antibiotics given after the intra-articular infusion?

The infusion of ADSCs and AECs were performed after 6 weeks from meniscectomy, while SVF was administered at the same day of meniscectomy. The infusion was performed without anesthesia and anti-inflammatory drugs. So, this information was given as follows: “Without anesthesia, the infusion of ADSCs and AECs were performed after 6 weeks from meniscectomy, while SVF was administered in the same day of meniscectomy.” (“Appendix A” section, page 13, lines 483-484). 

How was the intra-articular access site secured after the intervention? How were the animals kept, individually or in groups? How were the sheep euthanized 3 or 6 weeks after administration of ADSCs, AECs, SVF, or control saline?

The intra-articular site access was secured with suture after intervention and the animals were kept individually after intervention. After 3 and 6 months, sheep were pharmacologically euthanized with intravenous administration of 20 mL m-butamide, mebenzonium iodine, and tetracaine chloride Tanax® (Intervet-Italia S.r.l., Milan, Italy) under deep general anesthesia. The following sentences have been added: “The intra-articular site access was secured with suture and the animals were kept individually after intervention. After 3 and 6 months, sheep were pharmacologically euthanized with intravenous administration of 20 mL m-butamide, mebenzonium iodine, and tetracaine chloride Tanax® (Intervet-Italia S.r.l., Milan, Italy) under deep general anesthesia.” (“Appendix A” section, page 13, lines 485-488).

How were the samples taken?

The following sentence has been added: “Synovial fluid was harvested with a syringe, the joints were carefully opened, and the macroscopic appearance of tibial plateau and femoral condyles was evaluated. After macroscopic evaluation, the entire knee joint was harvested for histology, histomorphometry and immunohistochemistry.(“Appendix A” section, page 13, lines 489-492).

All this information is necessary for the reader to properly understand the presented research results. I agree, you can use references to describe the preparation of ADSCs, AECs, and SVFs, but the description of the animal experiment should be complete.

L 82 Were these commercial tests? Was the test specific for the sheep specimen? Please, provide sensitivity, intraassay CV, and interassay CV for each ELISA kit. What sample dilution was used? Was a biochemical, cytological, and bacteriological examination of the synovial fluid performed?

These are ELISA commercial tests specific for the sheep specimen. No dilution was used, and no biochemical, cytological and bacteriological examination of synovial fluid was performed.

For IL1beta the minimum detectable dose (MDD) is typically less than 3.9 pg/mL; Intra-Assay: CV<10%; Inter-Assay: CV<15%.

For IL6, the minimum detectable dose (MDD) is typically less than 0.7 pg/mL; Intra-Assay: CV<10%; Inter-Assay: CV<15%.

For TNFα, the minimum detectable dose (MDD) is typically less than 1.82 pg/mL; Intra-Assay: CV<10%; Inter-Assay: CV<15%.

For CTx2, the minimum detectable dose (MDD) is typically less than 39 pg/mL; Intra-Assay: CV<10%; Inter-Assay: CV<15%.

For PGE2, the minimum detectable dose (MDD) is typically less than 10 pg/mL; Intra-Assay: CV<10%; Inter-Assay: CV<15%.

So, the following modification was done as follows: “Inflammatory factors of the synovial fluid were quantified with ELISA kits, according to manufacturer indications (ABclonal Technology,Woburn, MA, USA): Interleukin 1β (IL1β) (minimum detectable dose-MDD- typically less than 3.9 pg/mL), Interleukin 6 (IL6) (MDD typically less than 0.7 pg/ml), Tumor necrosis factor α (TNFα) (MDD typically less than 1.82 pg/ml), Cross Linked C Telopeptide of Type II Collagen (CTx2) (MDD typically less than 39 pg/ml), and Prostaglandin E2 (PGE2) (MDD typically less than 10 pg/ml). All of ELISA kits have an intra-assay CV < 10% and inter-Assay CV < 15%.” (“Materials and methods” section, page 8, lines 284-290).

L 86 The table containing the DS scoring system should be provided.

Information about DS score have been reported in a previous article [17]. However, the following sentence has been now added to the Appendix A to better explain the score: “Gross evaluation of knee joint. Damage of the articular surfaces of each knee was assessed through a macroscopic score, evaluating the condition of the articular cartilage in the central area of the medial and lateral compartment of both tibial plateau and femoral condyles (4 quadrants) [17]. The final score was the sum of the 4 single quadrants (normal aspect of cartilage = 0; large erosions up to the subchondral bone = 16).” (“Appendix A” section, page 14, lines 508-513).

L 87 Were X-rays or CT scans of the knee joints taken to assess radiological signs of OA? Were such images taken prior to the procedure to rule out endogenous OA? Were goniometric measurements of the joint performed?

Unfortunately, these evaluations were not performed.

L 95 was primary H&E staining done?

Even if we recognize that H&E staining is important, however, in order to follow the procedures described in the winning project, from which the articles were obtained, we only performed the histological staining which allows us to highlight the quality and PG content of the cartilaginous matrix and the bone structure (Safranin O/Fast Green staining).

L 100 Expand the abbreviation (COLL II) when using it for the first time. Done.

L 102 Remove the additional dot. Done.

L 127 Figure 1. In the material and method section, there is nothing about H&E staining and classification of the histopathological results. All abbreviations used should be expanded in the caption of Figure 1.

I apologize for the mistake in the figure, but H&E staining should be replaced with Safranin O/Fast Green staining. For this reason, Figure 1, now Figure 3, has been modified. Also, in accordance with the reviewer's suggestions, the figure caption has been changed as follows: “…..OA = osteoarthritis; ADSCs = adipose derived mesenchymal stem cells; SVF = stromal vascular fraction; AECs = amniotic endothelial mesenchymal cells; NaCl = saline solution; IL1β = Interleukin 1 beta; IL6 = Interleukin 6; TNFα = Tumor necrosis factor alpha; CTx2 = Cross Linked C Telopeptide of Type II Collagen; PGE2 = Prostaglandin E2; DS = Damage Score; COLL II = Collagen II; CT = cartilage thickness; FI = fibrillation index; TH = Cartilage thickness; E = Young’s modulus; τ = relaxation time.” (“Materials and methods” section, page 9, lines 337-342).

L 156 In the results section, it is advisable to present the measurement values for each considered feature, not only for Young's modulus and relaxation time. Otherwise, describing their measurements in the materials and methods section is not justified.

As stated above, the purpose of the present study is to detect relationships between articular structure/composition and the mechanical properties of AC. Regarding the features retrieved by biochemistry of synovial fluid, gross evaluation of knee joint, histology, immunohistochemistry and histomorphometry of AC, their highlights have been reported in detail in a previously published article [17]. Nevertheless, aiming to clarify the bases on which this study is developed, we believe that reporting briefly the main experimental features of such analyses – as done in the various paragraphs of Materials and Methods, together with the peculiarities and the results of mechanical and Machine Learning analyses, not previously reported, is necessary and fair to understand properly the framework here reported.

L 171 Figure 2. All abbreviations used should be expanded in the caption of Figure 2 as well as the data shown in the plots. Do boxes represent the lower quartile, median, and upper quartile, whereas whiskers represent minimum and maximum values? Why are you using boxing plots since you declared in L 135 that data have verified normal distribution?

Caption of Figure 2, now Figure 1, has been revised accordingly. Regarding the meaning of boxplot representation, i) the central mark indicates the median, ii) the bottom and top edges of the box indicate the 25th and 75th percentiles, respectively, whereas iii) whiskers represent minimum and maximum values. According to the authors’ best knowledge, boxplot representation is appropriate also to summarize a series of ordinal or continuous data, as they display variation in samples of a statistical population without making any assumptions of the underlying statistical distribution.

L 174 There are no 1 symbol: p<0.05; 2 symbols: p<0.005; 3 symbols: p<0.0005 on a plot.

The symbol is § (p<0.05) and it is located in the part B of figure 1 where there is “6mo”. This indicates that there is significance between experimental times for all treatments, for τ.

The discussion also requires to be restructured. In its current form, no apparent logical order is used in comparing/contrasting the results obtained in the study with previous literature. The authors should attempt structuring the discussion following a "common standard format" that usually consists of the following points:

  1. One-sentence summary that highlights the most relevant results.
  2. A thorough discussion of each result obtained in relation to the corresponding study objective: was the tested hypothesis confirmed or not? Why? What previous evidence supports the specific result or not? It is critical to compare/contrast the result obtained with previous literature on the sheep first, then in other large animal models, and finally in human medicine.
  3. Statement of study limitations
  4. Future directions
  5. Conclusions

The “Discussion” section has now been modified according to the reviewer suggestions.

The conclusion section requires a complete rewrite so that it indicates the conclusions of the presented results and not general conclusions regarding the use of selected techniques in the large animal model of OA.

The “Conclusion” section has now been modified according to the reviewer suggestions. After implementing all the suggested changes, the results obtained might significantly differ from those currently reported in the manuscript. Thus, no specific comments are provided below, and a more thorough review of the discussion section will be provided once the manuscript is considered overall scientifically sound.

Reviewer 3 Report

Some questions can be addressed in the revision to improve the readability.

1.     The abstract gives a good overview, but it could benefit from more specific details about the machine learning approach used.

2.     It is important to include a clear introduction about the role of machine learning in this study and (if applicable) an overview about ML in this field in general. Given that the manuscript's title mentions "a Machine Learning Approach," the introduction should provide a brief explanation of how machine learning is used to analyze the data and reveal relationships between articular tissue properties and cartilage viscoelasticity. This inclusion will help readers understand the novel approach taken in this research and the potential benefits of using machine learning methods for the analysis.

3.     Could you please explain the potential underlying mechanisms for the observed relations between cartilage properties and synovial fluid biomarkers little bit more in detail in the discussion part?

4.     Could you elaborate on the potential implications of these findings for future studies or clinical applications? What are the limitations of the study, and how might they impact the generalizability of the results, in particular the ML approach? How do the results of this study compare to previous research in the field?

Author Response

We want to thank the reviewer for her/his thorough review of the article and her/his suggestions for improving the manuscript. Below are the changes, with line and page numbers, while in the text the changes are shown in red.

Many thanks from all the authors.

Some questions can be addressed in the revision to improve the readability.

1.The abstract gives a good overview, but it could benefit from more specific details about the machine learning approach used.

As suggested by the reviewer, we have added a summary of the machine learning methods used in the abstract: “After performing an initial analysis to evaluate the correlation and multicollinearity between the investigated variables, this study used machine learning (ML) models - Variable Selection Using Random Forests (VSURF) and Extremely Gradient Boosting (XGB) - to classify variables according to their importance and employ them for interpretation and prediction.” (“Abstract” section, page 1, lines 18-22).

  1. It is important to include a clear introduction about the role of machine learning in this study and (if applicable) an overview about ML in this field in general. Given that the manuscript's title mentions "a Machine Learning Approach," the introduction should provide a brief explanation of how machine learning is used to analyze the data and reveal relationships between articular tissue properties and cartilage viscoelasticity. This inclusion will help readers understand the novel approach taken in this research and the potential benefits of using machine learning methods for the analysis.

Following the reviewer suggestion, a paragraph about Machine Learning has been added to the Introduction: “Data were analyzed using a machine learning (ML) approach to establish a prediction model that combines various histological and biochemical factors related to OA, relating the mechanical response of AC (Young’s modulus and relaxation time) to insights about the pathophysiology of joint. The combination of ML and biomedical research has resulted in significant advancement in the field of human health by utilizing expansive biomedical data and tackling the growing complexity of biological systems, thus surpassing the limitations of traditional research methodologies. This has transformed the analysis and interpretation of intricate biological data, leading to the detection of biomarkers, the prediction of diseases, and the customization of therapies [30]. Biomedical analysis traditionally uses group comparison tests and regression for predictions. Assumptions of normal distribution and homogeneity of variance might not hold, leading to errors and overlooking complex variable interactions, impacting accuracy [31]. ML regression models such as regression tree model are predictive modeling technique based on decision tree structures and aim to predict a continuous numeric value as the output. The input data is split into subsets based on different features and their values in a regression tree. Aiming to minimize the variability of the target variable within each segment, the algorithm recursively partitions the data into smaller segments. For each step, the algorithm selects the feature and the corresponding value that best splits the data, typically based on criteria such as mean squared error or variance reduction. The result is a tree-like structure in which each leaf node represents a predicted numeric value. For predicting a new data point, the tree follows the path along the input feature values, and the final output is the prediction at the leaf node [30].” (“Introduction” section, page 2, lines 93-96 and page 3, lines 97-114).

In addition, new references have been added: “30. Jiang, F.; Jiang, Y.; Zhi, H.; Dong, Y.; Li, H.; Ma, S.; Wang, Y.; Dong, Q.; Shen, H.; Wang, Y. Artificial intelligence in healthcare: past, present and future. Stroke Vasc Neurol 2017, 2, 230-243.

  1. Gelman, A., & Hill, J. Analytical methods for social research: Data analysis using regression and multilevel/hierarchical models. Cambridge University Press 2006.” (“References” section, page 15, lines 596-599).

3. Could you please explain the potential underlying mechanisms for the observed relations between cartilage properties and synovial fluid biomarkers little bit more in detail in the discussion part?

Considering the absence of relative literature about possible relationships between the mechanical properties of the AC and knee joint inflammatory condition, the potential underlying mechanisms for the observed relations can be suggested only by arguing about the eligibility of the retrieved evidence. We are aware that this point represents the main limitation of the study, as now highlighted in the last part of Discussion section: “Prior to summarize about the evidence achieved, few caveats must be highlighted. Considering both the absence of relative literature and the pilot implementation of the proposed approach, the meaning behind the relationships between AC functionality and synovial fluid inflammation is only suggested, in particular by arguing about possible connections. Moreover, and still relating to the previous point, a broad investigation should be carried out prior to generalize our findings; in this perspective, a starting point could be represented by studies based on small animals OA model, due to their availability and cost [52]. Despite the above mentioned limitations, the retrieved evidence should be taken into account aiming to develop a multidisciplinary framework to study OA features comprehensively. In this perspective, achieving information about AC features starting from synovial fluid analysis could be a game-changer approach in understanding the onset and progression of OA, especially considering the potential implications of retrieving a full knowledge about the condition of the knee articular tissues from a diagnostic tool employable since the onset of the early stages of degenerative pathologies.” (“Discussion” section, page 7, lines 259-273).

4. Could you elaborate on the potential implications of these findings for future studies or clinical applications? What are the limitations of the study, and how might they impact the generalizability of the results, in particular the ML approach? How do the results of this study compare to previous research in the field?

As reported in response to the previous concern, according to authors knowledge, no study investigated possible relationships between the mechanical properties of AC and knee joint inflammatory condition; consequently, it is not possible to compare the results achieved in this sense to the ones highlighted in previous research in the field. Nevertheless, we tried to improve the clarity and the quality of the manuscript by highlighting such a limitation, moreover revising Discussion and Conclusion sections according to the reviewers’ suggestions. 

Round 2

Reviewer 1 Report

The presentation of the study is now improved, and the paper can be accepted for publication.

Reviewer 2 Report

Dear authors,

I recognize the effort to improve the quality of the manuscript. It has largely improved, and now I consider it acceptable. Detailed references to your previous publications and the addition of necessary details of the experimental protocol greatly improved the reader's reception of this article. Congratulations on a job well done.

Best regards